# Prevalence of Intimate Partner Violence, Substance Use Disorders and Depression among Incarcerated Women in Lima, Perú

**DOI:** 10.3390/ijerph182111134

**Published:** 2021-10-23

**Authors:** Elena Cyrus, Jorge Sanchez, Purnima Madhivanan, Javier R. Lama, Andrea Cornejo Bazo, Javier Valencia, Segundo R. Leon, Manuel Villaran, Panagiotis Vagenas, Michael Sciaudone, David Vu, Makella S. Coudray, Frederick L. Atice

**Affiliations:** 1Department of Population Health Sciences, College of Medicine, University of Central Florida (UCF), Orlando, FL 32827, USA; dvu140@knights.ucf.edu (D.V.); makella.coudray@ucf.edu (M.S.C.); 2School of Public Health, Yale University, New Haven, CT 06520, USA; frederick.altice@yale.edu; 3Centro de Investigaciones Tecnológicas, Biomédicas y Medioambientales, Callao 07006, Peru; jsanchez@impactaperu.org; 4Department of Health Promotion Sciences, Mel & Enid Zuckerman College of Public Health, University of Arizona, Tucson, AZ 85724, USA; pmadhivanan@arizona.edu; 5Public Health Research Institute of India, Mysore 560020, Karnataka, India; 6Asociación Civil Impacta Salud y Educación, Lima 15603, Peru; jrlama@impactaperu.org (J.R.L.); jvalencia@impactaperu.org (J.V.); manuelvillaran@hotmail.com (M.V.); 7International Degrees Department, Universidad Peruana de Ciencias Aplicadas, Lima 15023, Peru; pcukacor@upc.edu.pe; 8Office of Research and Technology Transfer, Universidad Privada San Juan Bautista, Chorrillos 15067, Peru; SEGUNDO.LEON@upsjb.edu.pe; 9Berkeley Research Development Office, University of California, Berkeley, CA 94704, USA; panagiotis.vagenas@yale.edu; 10Division of Infectious Diseases, University of North Carolina at Chapel Hill, Chapel Hill, NC 27599, USA; michael.sciaudone@unchealth.unc.edu; 11Tulane School of Medicine, New Orleans, LA 70112, USA

**Keywords:** women, incarceration, violence, substance-use, depression, Perú

## Abstract

Background: Globally, there is evidence supporting the co-occurrence of intimate partner violence (IPV), substance use disorders (SUD) and mental health disorders among women in prisons, however, there is limited research investigating these domains in the Andean region where rates of female incarceration have increased. The study objective was to explore the prevalence of IPV, SUD and depression among incarcerated women in a Peruvian prison and explore associations among these variables and related correlates. Methods: 249 incarcerated women responded to a questionnaire about IPV, substance use, depression, and sexual behavior, and were screened for HIV/sexually transmitted diseases (STDs). Univariate analysis and logistic regression were used to estimate relative risk and the influence of substance use and depression on IPV rates. Results: Twelve months prior to incarceration, of the women with sexual partners pre-incarceration (n = 212), 69.3% experienced threats of violence, 61.4% experienced ≥1 acts of physical violence, and 28.3% reported ≥1 act of sexual aggression. Pre-incarceration, 68.1% of drug-using women had a SUD, and 61.7% of those who consumed alcohol reported hazardous/harmful drinking. There were 20 (8.0%) HIV/STD cases; and 67.5% of the women reported depressive symptoms. Compared to women with no experiences of physical violence, a greater proportion of women who experienced least l violent act had depressive symptoms and engaged in sex work pre-incarceration. Depression was associated with physical violence (adjusted relative risk = 1.35, 95% confidence interval: 1.14–1.58). Recommendations: The findings provide evidence of a syndemic of IPV, substance abuse and depression among incarcerated women in a Peruvian prison. To help guide policy makers, further research is needed to determine if this is indicative of trends for other at-risk women in the region, and viable options to treat these women during incarceration to prevent recidivism and other long-term negative sequalae.

## 1. Introduction

Violence against women is a pervasive public health problem affecting women worldwide—particularly those from lower and middle income countries [1]. Globally, rates of violence have been consistently increasing [2], and the impact of that increase is more pronounced among victims of sexual violence in countries with a low human development index (HDI) [3]. However, even in countries with a higher HDI, such as the US, socio-economic disparities drive the negative impact of intimate partner violence (IPV) on substance use, mental health and sexual risk among lower income women [4]. The effect is amplified among incarcerated women where there is often a history of violence and trauma which is known to have an impact on mental health [5,6,7].

Reported rates of physical and/or sexual intimate partner violence in the Americas is the second highest regional estimate globally at approximately 30%, indicating an epidemic of violence against women in this region [1,8]. In a regional comparative study of 12 Latin American countries, Perú had the third highest documented rate of violence against women, including severe physical and sexual violence [8,9].

In Perú, between 2001 and 2015, the prison inmate population increased almost three-fold from 26,394 to 75,379 [10] and female prisoner rates have more than doubled in the past 14 years. Over 50% of female convictions in Perú are related to illicit drug use, trafficking, and smuggling [10], which may result from the cocaine drug trade that is pervasive in the region [11]. Perú is one of the largest producers of cocaine in the region [12,13], which is reflected in the surging rates of drug-related crimes [14]. The increase in the female incarcerated population can also be attributed to disparities in the judicial system whereby women who are charged with narcotics related crimes are typically incarcerated during the judicial process. After sentencing occurs, there is a further distinction because women associated with drug-cartels are less likely to have the opportunity for early release, probation or other benefits, as compared to women who may have operated “independently” as a drug trafficker [15].

There is evidence to support an association between these trends and the plight of lower-income Peruvian women, who may opt to engage in the underground cocaine economy to alleviate economic strain and related stress [14,16]. Women’s increasing participation in drug related crimes may stem from the vulnerabilities and risk they face because of less economic opportunity and higher rates of poverty compared to Peruvian men [17,18]. Although poverty rates have decreased in Peru, in 2016, the female unemployment rate was higher than the national average, and more than 20% higher than the male unemployment rate (5.6% vs. 4.4%) [18].

Elevated rates of physical and/or sexual violence are associated with higher rates of depression, high-risk sexual behavior, and substance-use in women [19]. Specifically, violence against women has been shown to double the probabilities of women experiencing depression and alcohol use disorders. Women who experience violence are nearly twice as likely to acquire HIV and sexually transmitted infections (STIs) such as syphilis, chlamydia, and gonnorhea [1,20]. These interacting factors of substance use, violence, and HIV/AIDS have been referred to as the SAVA syndemic [21]. In other research with other high-risk female populations, SAVA syndemic theory has been used has a guiding framework to examine health outcomes such as mental health, and will also be used in this study [22,23].

Despite the potential interaction of these three factors [intimate partner violence, substance use (alcohol and illicit drugs), and mental health/depression] among the rising incarcerated female population in Peru and the Andean region there is a dearth of scientific research exploring this syndemic. Although there is a more extensive body of work focusing on similar research in Brazil and other South American countries [24,25], to our knowledge, the present study is one of the first attempts to understand the association of intimate partner violence, alcohol use, and illicit drug use among incarcerated women in this region.

To partially address this scientific gap and provide evidence to inform the development of interventions that can curtail the growth of this syndemic, the purpose of the present study was to firstly examine the prevalence of IPV among female prisoners in Lima, Perú, and, secondly to assess the relationship between IPV, substance use, depression and indicated correlates. This study will add to the literature on an ongoing public health problem that is not only relevant for Perú, but other countries in the Andean region with similar social determinants of health for women.

## 2. Materials and Methods

Between May and July 2015, there were approximately 779 female inmates in Santa Monica Prison (Chorillos, Lima, Perú). During this period, the study team conducted weekly information sessions in Spanish on general women’s reproductive health and invited inmates to participate in the IPV study. Women who were interested were screened for eligibility to participate in the study. To be eligible, participants had to be ≥18 years of age, willing to complete the computer assisted self-interview (CASI) and provide venous blood and vaginal swabs for HIV/STI testing. The study visit was approximately one hour long (30–45 min for the interview and 15 min for lab work), and participants were provided with toiletry items for their participation in the study.

All interested and eligible participants underwent informed consent process before being enrolled into the study. Prior to enrollment, participants were provided the opportunity to ask questions about the study and the informed consent. The verbal informed consent process was completed in a predesignated private space in the prison facility—with one prison staff standing outside of the room. Of the 450 women who attended the informational sessions, 249 (55.3%) inmates were enrolled into the study. Once enrolled in the study, participants were assigned a unique study identification number (ID)and the study ID. Instead of the participant’s personal information, the data was coded with a unique identifier. Aside from the principal investigator, study team members who facilitated the consent process, CASI, and biological specimen collection were Peruvian nationals and spoke fluent Spanish. No personal identifiers were used during study collection, and the unique study IDs were used for all reporting procedures.

CASI was administered in a private room provided by the prison administration, and the questionnaire programmed in CASI was in Spanish. The questionnaires in CASI that were not already validated in Spanish, were vetted with the research team and the community advisory board of La Asociación Civil Impacta Salud y Educación (IMPACTA). To facilitate openness and honesty of respondents, prison guards or other prison personnel were not present during the CASI administration. All study procedures were reviewed and approved by two university institutional review boards in the U.S. (Yale University and Florida International University), and the ethics review board for IMPACTA in Lima.

### 2.1. Variables

Variables were selected a priori based on the theoretical framework of substance use, violence, and HIV/AIDS (SAVA) syndemic [21], particularly relevant for young women where there may be a synergistic effect of gender based violence, high-risk sexual behavior, and alcohol and illicit drug use.

#### 2.1.1. Socio-Demographic and Sexual Risk Factors

Socio-demographic information collected included age of participant, number of live births, country of birth, and level of education (no schooling, partial or full completion of primary school, secondary school and beyond). Sexual risk behavior prior to incarceration included questions on HIV screening, engagement in sex work, and condom use. Participants were also asked about their incarceration time and the number of minor children under the age of three that resided with them in the prison.

#### 2.1.2. Intimate Partner Violence

The Severity of Violence Against Women Scale (SAVAWS) [26] was used to assess women’s experience with IPV before being incarcerated. SAVAWS is a validated 47-item scale (Appendix A) with subscales on threats of violence (items 1–20), acts of physical violence (items 21–41), and sexual aggression (items 42–46). Thirty-seven women reported having no intimate partner prior to incarceration, leaving 212 (85.1%) women to complete the violence scale.

For every item on the scale, participants responded 1 = never, 2 = once, 3 = a few times, or 4 = many times. Scores for threats of physical violence (range 19–76), physical violence (range 27–108) and sexual aggression (6–24) were summed to obtain three total scores for each subcategory of IPV. In descriptive analysis, for participants who reported experiencing a threat or act of violence, summed scores were dichotomized into low/moderate threat and abuse, and strong/severe threat and abuse. Using precedent in the literature from previous research using the SAVAWS scale, a cut-off of 25 was used for threats of violence, and 27 was used for physical violence [27,28]. The Cronbach’s alpha reliability coefficients of the SAVAWS subscales were 0.95, 0.96, and 0.86 for threats of violence, acts of physical violence, and sexual aggression, respectively.

#### 2.1.3. Substance Use (Illicit Drug and Alcohol Use)

The short version of the drug abuse screening test (DAST-10) was used in the study [29], and data for hazardous drinking were assessed using the 10-item Alcohol Use Identification test (AUDIT) scale [30]. For univariate analysis, dichotomous variables were constructed from the summed DAST-10 and AUDIT scores by using a total score of ≥6 as ‘substantial/severe’, and <6 as ‘non-severe’ for DAST-10, and, using a total score of ≥7 as ‘hazardous’ drinking, and <7 as ‘non-hazardous’ drinking as defined per AUDIT criteria for women.

#### 2.1.4. Depression

The abbreviated version of the Center for Epidemiologic Studies Depression Scale (CESD-10) was used to assess depression [31]. For univariate analysis, a dichotomous variable was constructed from the summed scores by using a total score of ≥10 as having significant depressive symptoms, and <10 as absence of any significant depressive symptoms.

#### 2.1.5. Sexually Transmitted Infections

Women were evaluated for human immunodeficiency virus (HIV), syphilis, *Neisseria gonorrhoeae*, and *Chlamydia trachomatis*. Venous blood samples and vaginal swabs were collected in the prison infirmary and transported to the Universidad Peruana Cayetano Heredia laboratory for processing. Alere Determine™ HIV ½ test (Alere Diagnostics, Chiba-ken 270-2214, Japan) was used for HIV rapid blood testing, and positive HIV rapid test results were confirmed by Western blot. Syphilis was evaluated using rapid plasma reagin (RPR) with microhemaglutination assay and treponema pallidum (MH-TP) for confirmation. *Chlamydia trachomatis* and *Neisseria gonorrhoeae* were identified using Aptima Combo2 CT/NG TMA test (Hologic, San Diego, CA, USA). Confirmed positive HIV/STI cases were notified within two weeks of the confirmation of their results. The study coordinator provided positive test results and associated post-test counseling to the inmates. The results were also provided to prison personnel to allow for referral for appropriate care and treatment.

### 2.2. Analysis

Data were checked for normality distribution and outliers. While there were no significant outliers, the data for the IPV variables were heavily right skewed. After unsuccessful attempts of square root and log transformation of the data, a decision was made to dichotomize the data (ever experience an episode of IPV versus never experience an episode of IPV prior to incarceration) for regression analysis.

To inform development of statistical models that would test associations between the IPV domains and correlates, descriptive univariate analyses were conducted on all demographic indicators and outcome violence variables (n = 249). Thirty-seven women reported not having intimate partners before entering prison, decreasing the sample size to 212. Adhering to the SAVA theoretical model, chi-square analysis was used to test associations between the outcome variables (threat of abuse, physical abuse, sexual aggression, and total IPV) substance use and HIV/STI status, as well as depression, engagement in sex work, and use of condoms prior to incarceration. These variables were also used to estimate relative risk and associated 95% confidence intervals (95% OR) for crude and adjusted models for threats of violence, physical violence, sexual aggression and total IPV. Although sex work was found to be significant during bivariate analysis, it was ultimately excluded from final regression models due to lack of model fit. Adjusted models included age and level of education as covariates. A value of *p* <0.05 was considered significant. All analysis was performed using SPSS 24.0 (IBM, New York, USA) [32].

## 3. Results

The sociodemographic characteristics of the study sample are presented in Table 1. Among the women who participated in the weekly reproductive health informational session and volunteered to respond to the questionnaire, the majority of the women (93.5%) were Peruvian, followed by Mexican women (3.6%); the remainder of the sample (2.8%) comprised women with the following national origins: Venezuela, Spain, Ecuador, Paraguay and South Africa. Only 33 women (13.2%) indicated they had prison sentences greater than five years. The median age of the women was 37 years [range 18 to 70 years]. Six women (2.6%) were pregnant in prison, 18 women (7.2%) had children under the age of three residing with them in prison, and six of these women (3%) were breastfeeding. At the time the study was conducted, just over half of the women interviewed (57.4%) had been in the prison for five years or less at the time of their study visit, 76 women (30.5%) had been in the prison between 5 and 9 years. The majority of women (78.7%) had completed at least a high school degree or some college.

### 3.1. Depression and Intimate Partner Violence

More than two-thirds (n = 168, 67.5%) of the participants had depressive symptoms in accordance with the CESD-10 criteria. Of the 212 women who completed the violence questionnaire, 147 (69.3%) women reported experiencing at least one moderate or strong threat of violence, and 153 women (72%) reported experiencing at least one act of physical violence 12 months prior to incarceration. Most of the women (n = 193, 91%) experienced at least one type of acts of sexual aggression prior to incarceration (see Table 2).

### 3.2. Substance Use

Pre-incarceration, of the 66 women reporting drug use, 68% had a substance use disorder (SUD), and of the 212 women who drank alcohol, 61.7% had an alcohol use disorder (AUD) (Table 1). In prison, 18 women (7.2%) reported alcohol use and 24 women (9.6%) reported illicit drug use. The majority of alcohol used in the prison was liquor (chicha) prepared in-house illicitly by the inmates. Marijuana and cocaine were the primary substances used before and during incarceration. Five women reported using ‘pills’ for sleeping or to deal with depression; however, there was no clarification if the pills were over-the-counter medicine or prescribed by a healthcare provider.

### 3.3. Correlates of HIV/STD and Sexual Risk

In prison, 37 women (14.9%) reported having conjugal visits. There were four (1.6%) HIV cases, 10 (4.0%) chlamydia cases, four (1.6%) syphilis cases, and two (0.8%) gonorrhea cases (data not presented in tables) among the sample. Only 18.8% of the women reported condom use with their intimate partners prior to entering prison (76.5% reported using condoms sometimes and 23.5% used them always). Twenty-eight women (11.2%) reported engaging in sex work prior to incarceration and eight of them reported not using condoms during those sexual transactions. Three women who engaged in sex work were living with HIV.

### 3.4. Bivariate Analysis and Logistic Regression

In bivariate analysis (Table 3), several trends emerged according to category of violence. A greater proportion of women who experienced physical violence had depressive symptoms (81.6% vs. 59.9%, *p* < 0.0001), and engaged in sex work prior to being incarcerated (14.9% vs. 9.3%, *p* = 0.18). A greater proportion of women wo experienced sexual aggression had depressive symptoms, engaged in substance use, was HIV infected or had an STI, and, engaged in sex work prior to incarceration.

This association between violence and depression was consistent in the multivariable analysis (Table 4), where depression was significantly associated with physical violence (aRR = 1.35 95%CI 1.14–1.58) after adjusting for education level and age.

Post-hoc bivariate analysis between drug use and hazardous drinking by engagement in sex work (data not presented in tables), illustrated that compared to women who did not engage sex work, a higher percentage of women involved in sex work reported illicit drug use (60.7% vs. 24.0%, *p* < 0.001) and hazardous drinking (71.4% vs. 43.9%, *p* = 0.005). However, there was no difference in depressive symptoms by sex work status, perhaps because over two-thirds of the sample population reported depressive symptoms and there was not enough variance in that outcome (depressive symptoms) to detect differences by sex work status or another demographic factor.

## 4. Discussion

In our sample, both threats and actual physical violence were highly prevalent—with more than a third of the women reporting strong threats or severe abuse before they were incarcerated. The key findings of this study were the high rates of threats of violence and physical violence regardless of substance use, suggesting a persistent societal ill of pervasive violence against women. Additionally, the majority of women reported substance use disorders including hazardous/harmful drinking pre-incarceration and had depressive symptoms. There was an association between threats/experiences of violence and depression, which partially explains the similar violence and depression rates among the sample (69.3% violence and 67.5% depression). These findings are consistent with other female incarcerated populations that shows evidence of elevated levels of violence and depression among incarcerated populations compared to the general population, especially among repeat offenders [33,34]; however, because of global trend overall of rising female incarceration, this presents an urgent public health problem to be addressed [35].

Mimicking global trends protesting the culture of violence against women, the Peruvian government recognizes the permeating culture of violence against women in Peruvian society and has made strong efforts to address the issue. The government has implemented programs to engender societal attitude shifts regarding women, including acknowledgment and celebration of the “Dia de la Mujer” (Womans’ Day) and supporting local campaigns with social marketing advocating for the rights of women and other vulnerable populations [36]. In 1996, the Ministerio de la Mujer was established to reinforce the rights of Peruvian women and children. Despite these efforts, economic disparity is borne heavily by Peruvian women, and traditional *machismo* (male dominance) norms persist, stemming social progression for women’s rights [14]. With the geopolitical landscape remaining unstable, vulnerable women will be more likely to resort to the informal sector including survival sex work and the preexisting narco subculture in Peru and the region [37].

For ethical reasons, and due to the risk of compromising participants’ confidentiality, the present study did not directly explore the relation between simultaneous growth of cocaine drug trade and female incarceration rates; however, some logical assumptions about this relationship can be made based on the descriptive data analysis and other background information. Firstly, in Peru, women are more likely than men to be incarcerated for drug related crimes; secondly, Peruvian policy regarding illicit drugs typically administers sentences less than 15 years for minor drug related crimes, such as smuggling or low-level distribution/selling of drugs. Sentences longer than 15 years are reserved for individuals with heavier involvement in the drug trade such as production and wide-scale international distribution [14]. In our sample, most women had an incarceration time less than 15 years. Thirdly, rates of pre-incarceration SUDs in our population were almost 10-fold the rates of the general female population [38]. Thus, we deduced that a considerable proportion of women may have been serving sentences for drug related crimes (e.g., illicit drug use; drug smuggling or low-level distribution) due to (a) background national statistics regarding incarceration of Peruvian women and (b) large proportion of the sample having short-term sentences and (c) high SUD prevalence in the study sample.

In this study, despite most of the women being educated, for unknown reasons, they elected to engage in drug trade for economic survival. Reasons for drug-trade involvement are not absolutely known for this study sample, but women may choose to engage in these activities to combat their low socio-economic status, or, conversely are coerced to participate by dominant male figures (partners or otherwise) in their lives [14]. Both illicit drug use and/or involvement in the drug trade can increase chances of these women being victims of violence or threats of violence, and ultimately can increase their overall vulnerability.

High rates of depressive symptoms in this population of women, some of whom were mothers, is consistent with the literature other female prison populations in the Americas, and with the general female Peruvian population [17,39]. However, further investigation is needed to determine if the women suffered depressive symptoms prior to incarceration, or, if being incarcerated initiated depressive symptoms, or if there is a bidirectional relationship. The literature supports an association between depression/mental health and poverty for women who are at poverty level but not incarcerated [39]. Depression prior to incarceration can contribute to illicit drug use or drug trade involvement, thereby increasing odds of incarceration [40]. Additionally, rates of depression can increase post-incarceration and increase likelihood of engagement in substance use, victimization from IPV, and odds of recidivism [41,42].

The limited sample size precluded analysis to test the association of HIV/STIs to IPV or substance use; however, HIV/STI prevalence in the prison population is higher than the female Peruvian general population and other high-risk groups such as female sex workers [43,44]. Only three women who tested positive for an HIV/STI reported engaging in commercial sex-work prior to incarceration; however, engagement in sex work may have been underreported due to social desirability bias. In this population, rates of syphilis were double that of female sex workers in Perú (1.6% vs. 0.8%) and four times that of women in the general Peruvian population (1.6% vs. 0.4%); Gonorrhea was eight times that of women in the general population (0.8% vs. 0.1%) [43].

### Limitations

The sample size was not large enough to detect all possible associations, and there was no opportunity to expand the sample size and increase power to detect associations with less magnitude. There may have been some recall bias, as women were asked to recollect events of violence 12 months prior to incarceration. Selection bias may have occurred as women who volunteered to participate in the study might have been more likely to be share about their IPV experiences compared to women who did not want to participate in the study. Furthermore, social desirability leading to the underreporting of sensitive and highly stigmatized issues related to illicit drug use and sexual behavior may have occurred. With the exception of selection bias, all other biases would have underestimated the magnitude of effect and minimized the gravity of this syndemic among incarcerated women. Finally, no inference can be made about the temporality of the relationship between IPV, substance use, and depression due to the cross-sectional nature of the study.

## 5. Conclusions

This study provides empirical evidence of a public health problem for women living in Perú, other Andean countries, and other global regions. This study population had elevated and overlapping levels of IPV, substance use, depression, and HIV/STIs. This population may benefit from proven interventions that have been applied in other correctional settings—that can be implemented during their prison time to address these issues [45]. Most of the female prisoners were in their childbearing years and serving relatively short sentences—facing imminent release into their communities. However, if they are facing the same risk factors that existed pre-incarceration, the likelihood of reengagement in high-risk behaviors post-incarceration will prolong or increase their vulnerability and increase their likelihood of recidivism. Higher rates of recidivism among repeat offenders is associated with a higher likelihood of experienced violence and depression. Moreover, a substantial portion of these women were mothers, or of reproductive age, which suggests the potential of an indirect negative effect on their children, creating a cyclical multi-generational pattern of substance use, violence, and incarceration among mothers and children [46,47,48].

Future multi-site longitudinal studies are needed to evaluate the relationships and factors this study did not have the power to assess. Successful interventions that have been acceptable and efficacious in other female prisoner populations globally, which address intimate partner violence, substance use, and/or depression [49,50,51], can also be considered for incarcerated women in Perú and South America. To help guide policy makers, further research is needed to determine if this is indicative of trends for other at-risk women in the region, and viable options to treat these women during incarceration to prevent recidivism and other long-term negative sequalae.

## Figures and Tables

**Table 1 ijerph-18-11134-t001:** Sample characteristics of female Peruvian inmates in Santa Monica Prison, Lima, Perú (n = 249); May–July 2015.

Median Age in Years	37	(Range 18–70)
Average number of live births	m = 2.7	(range 0–11)
COUNTRY OF ORIGIN	**n**	**%**
Perú	293	93.50%
Mexico	9	3.60%
Other	7	2.80%
EDUCATION		
No school	3	1.20%
Primary school	49	19.10%
Secondary school or higher	196	78.70%
SEXUAL RISK BEHAVIOR		
*Ever had HIV test*		
Yes	205	82.30%
No	4	1.60%
Missing	40	16.00%
*Commercial Sex Work*		
Yes	28	11.20%
No	221	88.80%
*Condom use before prison*		
Yes	47	18.80%
Sometimes	36	76.50%
Always	11	23.50%
No	149	59.80%
Missing	53	21.20%
PRISON SENTENCING		
*Incarceration Time*		
0–4 years	143	57.40%
5–9 years	76	30.60%
10–15 years	30	12.00%
*Children residing with inmates*		
Yes	18	7.20%
No	231	92.80%
DEPRESSION ^a^		
Depressive symptoms	168	67.50%
None/Minimal depressive symptoms	81	32.50%
DRUG USE (DAST-10) ^b^		
No drug use	183	73.50%
Ever used drugs	66	26.50%
No substance use disorder	21	31.90%
Substance use disorder	45	68.10%
ALCOHOL USE (AUDIT) ^c^		
No alcohol use	37	17.40%
Lower risk drinking	50	23.50%
Problematic drinking	131	61.70%
Missing	31	14.60%
PRISON SUBSTANCE USE		
*Drug Use*		
Yes	24	9.60%
No	225	90.30%
*Type of drugs*		
Marijuana	14	58.30%
Cocaine	12	50.00%
Pills ^d^	5	20.80%
*Alcohol Use*		
Yes	18	7.20%
No	231	92.70%
*Type of alcohol among*		
Home made	12	66.70%
Spirits	7	38.80%
Beer and other	4	22.20%
Other	7	38.80%

^a^ Scores ≥ 10 categorized as depressive symptoms per CESD-10 criteria. ^b^ DAST-10 (Drug Abuse Screening Test-10) classification of substance use disorders as scores ≥ 6. ^c^ AUDIT (Alcohol Use Disorder Identification Test) classification of problematic drinking defined as scores ≥ 7. ^d^ For treatment of depression or sleep disorders.

**Table 2 ijerph-18-11134-t002:** Frequency of Intimate Partner Violence (IPV) by categories among incarcerated Peruvian Women (N = 212); May–July 2015.

	n	%
*Threats of physical violence 12 months prior entering prison*		
No threat	65	30.6
Low/Moderate threat	73	34.4
Strong threat	74	34.9
*Physical violence 12 months prior entering prison*		
No abuse	59	27.8
Low/Moderate abuse	82	38.7
Severe abuse	71	33.5
*Sexual aggression 12 months prior entering prison (non-exclusive) **		
Demanded sex	60	28.3
Forced oral sex	26	12.3
Forced sexual intercourse	42	19.8
Rape	39	18.4
Forced anal sex	18	8.5
Forced to use object in a sexual way	8	3.8

* Note: These numbers do not total 100%.

**Table 3 ijerph-18-11134-t003:** Bivariate associations of selected risk factors for SAVA syndemic (n = 249).

	Threat of Violence			Physical Violence			Sexual Aggression			Total IPV		
	No	Yes	*p*	No	Yes	*p*	No	Yes	*p*	No	Yes	*p*
Depression	68.1%	65.3%	0.73	**59.9%**	**81.6%**	**<0.01 ***	66.3%	71.2%	0.48	70.2%	66.7%	0.89
Hazardous/Harmful alcohol use pre-incarceration	72.0%	64.3%	0.47	64.5%	64.8%	0.97	62.8%	70.5%	0.35	63.3%	64.9%	0.87
Ever used illicit drugs pre-incarceration	40.4%	28.0%	0.12	27.2%	29.9%	0.65	26.8%	32.2%	0.42	21.7%	30.2%	0.18
HIV/STI	8.5%	8.5%	0.99	8.0%	6.9%	0.75	6.8%	10.2%	0.40	5.3%	8.3%	0.44
Commercial sex work pre-incarceration	17.0%	9.3%	0.16	9.3%	14.9%	0.18	**8.9%**	**18.6%**	**0.03 ***	10.5%	11.5%	0.84
Condom Use pre-incarceration	26.5%	26.3%	0.98	26.8%	19.2%	0.23	25.5%	19.6%	0.39	30.6%	22.5%	0.30

**bold** and * significant at α < 0.05.

**Table 4 ijerph-18-11134-t004:** Results for linear regression models examining associations between violence and substance use among female Peruvian inmates with partners (n = 212).

	Threats of Violence		Physical Violence		Sexual Aggression		Total IPV	
	RR (95% CI)	aRR (95% CI)	RR (95% CI)	aRR (95% CI)	RR (95% CI)	aRR (95% CI)	RR (95% CI)	aRR (95% CI)
Age in years	-	1.01 (0.93–1.03)	-	1.02 (0.99–1.05)	-	1.01 (0.94–1.02)	-	1.01 (0.98–1.02)
Education	-	0.94 (0.71–1.23)	-	1.06 (0.98–1.16)	-	0.99 (0.91–1.06)	-	1.05 (0.98–1.12)
Depression	0.91 (0.54–1.49)	0.90 (0.54–1.49)	**1.39 * (1.17–1.59)**	**1.35 * (1.14–1.58)**	1.05 (0.91–1.21)	0.93 (0.93–1.25)	1.03 (0.86–1.19)	1.03 (0.90–1.25)
AUDIT (alcohol)	1.33 (0.61–2.94)	1.20 (0.54–2.70)	0.99 (0.78–1.28)	1.02 (0.80–1.29)	0.92 (0.78–1.08)	0.91 (0.78–1.07)	0.99 (0.89–1.16)	1.01 (0.88–1.16)
DAST-10 (drugs)	0.74 (0.42–1.09)	0.69 (0.41–1.13)	1.04 (0.85–1.23)	1.04 (0.84–1.28)	1.06 (0.91–1.25)	1.12 (0..95–1.31)	0.90 (0.80–1.03)	0.92 (0.80–1.06)

**bold** and * significant at α < 0.05.

## Data Availability

The data presented in this study are available on request from the corresponding author. The data are not publicly available due to privacy.

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
