# Peer review of "Prevalence of Intimate Partner Violence, Substance Use Disorders and Depression among Incarcerated Women in Lima, Perú"

_ijerph, 2021, doi:10.3390/ijerph182111134_

Round 1
Reviewer 1 Report
Women Without Liberty: Correlates of Intimate Partner Violence
Among Female Prisoners in Lima, Peru
(IJERPH-1364101)
The study addressed the relationship between intimate partner violence and “substance abuse” among incarcerated female prisoners in Lima, Peru. This research project had an opportunity to shed light on a pervasive issue that is prevalent among abused female prisoners. Unfortunately, I felt that the way the article was presented did not make the case based on the proposed research objective(s). An important concern surrounds the development of the background section, the analytic approach, and the way the information was presented. Furthermore, the researchers did not adequately provide an explanation of some of the study findings. Below I offer some suggestions for the author(s) to consider.
Abstract
- To begin, I felt that the title of the paper did not really capture the essence of the paper. Generally, the abstract is written well by highlighting the purpose, methodology, and results.
Introduction and Literature Review
- The authors provided good coverage of the epidemiology of intimate partner violence
- The author(s) might consider being more specific about the type of violence (i.e. IPV) related to depression, high-risk behavior and substance abuse
- The author(s) could start a new sentence regarding “women experiencing violence are twice as likely to acquire HIV………””
- The background could be better developed surrounding incarcerated IPV women.
- Although it is commendable that the author(s) provided context, at times it seems like the authors are addressing separate issues—whether it be IPV, incarceration, or substance abuse. Is the study on IPV, substance abuse or incarceration?
- The research objective could be clarified. The author(s) may consider being more explicit about the focus of the paper
- The theoretical framework could be illuminated and discussed more
Method
- If IPV was the focus of the study, how were participants screened into the study? It seems the screening was based on other aspects of the study (i.e., reproductive health)
- How are the sociodemographic factors in the study measured?
- Explain/ elaborate more on the cut-off points for some of the measures used—though references were given
- The analysis information could be better organized
Results
- Some of the information provided in-text were not reflected in the table and vice versa
- The results could be better organized and centered around the research objectives
- Why discuss findings that were not statistically significant, unless there is a reason or rationale for it
Discussion/Conclusions
- The author(s) could do a better job highlighting the substantive findings of the study and providing some explanation
- I am unable to grasp why the issue of Covid was included in the study
- Some findings seem to be reported independent of IPV—which I am assuming is the focus
- The author(s) may want to exercise better care during the submission process. It seems some valid suggestions from previous reviews were included---“Authors should discuss the results and how they can be interpreted from the perspective of previous studies and of the working hypothesis. The findings and their implications should be discussed in the broadest context possible. Future research directions may be highlighted”
- Other limitations were not addressed
Author Response
Attached is Word document of response to reviewer. Below is pasted version from Word document for convenience:
Manuscript IJERPH-1364101
Former Title: Women without liberty: correlates of intimate partner violence among female prisoners in Lima, Peru.
New Title: Prevalence of Intimate Partner Violence, Substance Use Disorders and Depression among Incarcerated Women in Lima, Perú
We would like to express our gratitude to the reviewers for such a detailed review that has strengthen the paper and improves the overall message of this manuscript which is the elevated risk and plight of these women due to multiple social and individual factors. All comments have been addressed, and details and our responses are outlined by section and reviewers below:
- TITLE
- Per suggestions from reviewers 1 and 2, the title has been changed to reflect the content and “capture the essence of the paper”. The new title is “Prevalence of Intimate Partner Violence, Substance Use Disorders and Depression among incarcerated women in Lima, Perú”
- BACKGROUND
- Per reviewer 1’s comments, the background has been further developed and includes more information around intimate partner violence, substance use and depression among incarcerated women.
- In response to reviewers 1 and 4, a theoretical framework has been elucidated
- MATERIALS AND METHODS
- In response to reviewers 1, 2, and 4, we have made the following changes to the materials and methods section:
- In second paragraph of the section, we expanded the description of the screening methods to explain how anonymity and confidentiality was protected
- We included language regarding procedures for HIV/STI testing and counseling
- In response to reviewers 1, 2, and 4, we have made the following changes to the materials and methods section:
- Further explanation and justification for the IPV cutoffs were provided and why binary logistic methods were employed
- RESULTS
- In response to reviewers 1, 2 and 3, we have corrected Table 1 and 2 formatting, and spelled out the acronyms “DAST” and “AUDIT”
- DISCUSSION
- Per reviewer 4’s suggestion, another paragraph has been added exploring the relationship between socioeconomic status and IPV among the study sample
- In response to reviewers 1 and 3, we:
- Removed the paragraph related to COVID-19
- Extended the discussion to discuss public health implications and future recommendations
- OTHER GENERAL CHANGES
- A funding section has been added
- Formatting for manuscript and references has been double checked and corrected
- In response to reviewer 4’s suggestion, term “intimate partner violence” is now used consistently throughout the manuscript

Reviewer 2 Report
Summary Review: Women without liberty: Correlates of intimate partner violence among female prisoners in Lima, Perú
|
1. Importance of article/Relevance and Appeal to national/international scholarly |
Excellent |
Good |
Moderate |
Poor |
|
|
2. Title of the study |
Excellent |
Good |
Moderate |
Poor |
|
|
3. Original and Independent Research |
Excellent |
Good |
Moderate |
Poor |
|
|
4. Presentation and readability |
Excellent |
Good |
Moderate |
Poor |
|
|
5. Statement of problem(s)/aim(s)/objective(s) |
Excellent |
Good |
Moderate |
Poor |
Not Available |
|
6. Literature review |
Excellent |
Good |
Moderate |
Poor |
|
|
7. Appropriateness of (if applicable) 7.1. Research plan and design 7.2. Data presentation/Discussion 7.3. Conclusion/Recommendations |
|
|
|||
|
Excellent |
Good |
Moderate |
Poor |
|
|
|
Excellent |
Good |
Moderate |
Poor |
|
|
|
Excellent |
Good |
Moderate |
Poor |
|
|
|
8. To what extent is the line of argumentation in the article clear, cohesive and logical? |
Excellent |
Good |
Moderate |
Poor |
|
|
9. Contribution to the theory |
Excellent |
Good |
Moderate |
Poor |
|
|
10. Contribution to practice |
Excellent |
Good |
Moderate |
Poor |
|
NOTE:
- The title needs to be review and correct relative to the body of the study.
- The Author(s) needs to do a lot of review work from INTRODUCTION down to the REFERENCES.
- This paper will be better situated if it follows the chronological order of writing an ABSTRACT that is: Introduction, Objectives, Data and Method, Findings/Results, policy implications, and Recommendations. More information needs to be added to the introduction for a better understanding of the study. The body of the work is totally different from the title. There has never been anything relating to alcohol, drug use, and substance in the title and this makes the free flow of thought/ reading to be difficult to synchronize.
- The analysis is not clear enough. Give a brief introduction to the binary logistic follow by the provision of the equation showing the understanding of the model.
- Kindly follow the suggestions given in the presentation of the table coupled with the headings.
- The reference format is not clear. If it is APA's six editions, kindly improve on it accordingly.

Author Response

(The authors gave the same response as above.)

Reviewer 3 Report
ABSTRACT
The study objective is well defined and identified both in the abstract and in the introduction.
INTRODUCTION:
The investigated topic is of growing scientific and social interest. The investigation is current.
MATERIALS, METHODS and RESULTS:
I think that the data collection process could have been improved if when asking about Intimate Partner Violence, they had been used a liker scale with five response options, so as not to polarize the results so much, instead of four. But this circumstance can no longer be changed.
In table 1, regarding its format, it is out of square, columns 2 and 3 are one row lower.
In table 2, data is repeated and is out of square.
Table 4 appears before table 3.
I think that table 4 should collect the significance of each variable.
DISCUSSION AND CONCLUSION:
The conclusions should be clearer and I would like to see some more conclusions, in relation to public policy action.
In the end, the authors introduce an analysis regarding the COVID disease, which I do not think is relevant, since the data with which the analysis has been made is from 2015, and the subject does not have a sufficient relationship.
Author Response

(The authors gave the same response as above.)

Reviewer 4 Report
ijerph-1364101
Women without liberty: Correlates of intimate partner violence among female prisoners in Lima, Perú.
The manuscript analyzes correlates of intimate partner violence in female prisoners in Lima. This is an interesting and little-known topic. However, the authors need to address some issues to improve the manuscript.
The objective of the study is defined differently throughout the text. Therefore, the authors should rewrite the objective where necessary, adapting it to the variables they have analyzed.
In page 2, they include HIV/AIDS when refer to the SAVA syndemic. However, it seems to be a less relevant problem for the purpose of the study in other paragraphs of the manuscript. They need to clarify why they consider that some of the correlates they measure are more important than others in their study.
In the abstract, the authors need to change interpersonal violence to intimate partner violence (these terms do not refer to the same type of violence).
A question of format: the punctuation marks that are now in front of the parentheses of the references, should be after the parentheses.
The authors indicate that drug trafficking is an important criminal activity in the region and that IPV is also prevalent. The authors should also consider that poverty can facilitate the visibility of IPV, and that some studies indicate that women get into drugs through the romantic relationships they establish with traffickers. This can facilitate the association between drug use/trafficking and IPV.
Materials and method: How were the anonymity and confidentiality of the participants’ responses ensured? what happened to the results of the venous blood and vaginal swabs for HIV / STI testing? Did the participants have any incentives for participating?
Please, specify the type of regression carried out in the Analysis section. Please, provide more information in the tables that describe the regressions. In a note under the tables, specify the meaning of the initials used.
Author Response

(The authors gave the same response as above.)

Round 2
Reviewer 1 Report
Women Without Liberty: Correlates of Intimate Partner Violence Among Female Prisoners in Lima, Peru
The study addressed the relationship between intimate partner violence and “substance abuse” among incarcerated female prisoners in Lima, Peru. The research has improved to some degree since the last submission. However, there are still some concerns mainly surrounding the background and results section. Below I offer some suggestions for the author(s) to consider.
Abstract
- While the abstract provided some understanding about the study, the findings section could be more focused on the study objective(s). I noticed that general descriptive information was included (i.e., age etc.) and I am not sure why.
Introduction and Literature Review
- Even though this section is improved from the last submission, it seems to be a bit disjointed in getting to the point about the study focus
- Clarify what is meant by “violence and trauma on mental health”
- I do believe the authors should stick to research objective(s), particularly if the focus is on mental health. Or, they need to indicate that they will explore other health-related conditions.
- I encourage the author(s) to be mindful of grammar in certain instances.
Method
- The author(s) should clarify whether they are assessing 12 months or lifetime violence
- How are the socio-demographic indicator measured?
Results
- The section presents a major concern for myself in terms of how it is organized and how it is presented. Again, this section should be centered around the research objective(s)
- Why discuss findings that were not statistically significant, unless there is a reason or rationale for it
- The table or table numbering is out of place
Discussion/Conclusions
- The author(s) could do a better job of highlighting the substantive findings of the study
- I would caution the author(s) about the term acceptable and normalization of violence
- While the study provides some indication of important global/societal issues, it is important for the author(s) to remind the readers about the population of individuals under analysis—incarcerated women who may have higher rates of these occurrences
- Some statements made should either be in the discussion section should either be in the limitation or results section.
- Other limitations were not addressed
- The conclusion could be stronger
